# Robust weak antilocalization due to spin-orbital entanglement in Dirac material Sr₃SnO

H. Nakamura[1,8 ✉], D. Huang [ID] [1], J. Merz[1], E. Khalaf[1,2], P. Ostrovsky[1,3], A. Yaresko [ID] [1], D. Samal[4,5] &
H. Takagi [ID] [1,6,7]

The presence of both inversion ($P$) and time-reversal ($T$) symmetries in solids leads to a double degeneracy of the electronic bands (Kramers degeneracy). By lifting the degeneracy, spin textures manifest themselves in momentum space, as in topological insulators or in strong Rashba materials. The existence of spin textures with Kramers degeneracy, however, is difficult to observe directly. Here, we use quantum interference measurements to provide evidence for the existence of hidden entanglement between spin and momentum in the antiperovskite-type Dirac material Sr₃SnO. We find robust weak antilocalization (WAL) independent of the position of $E_F$. The observed WAL is fitted using a single interference channel at low doping, which implies that the different Dirac valleys are mixed by disorder. Notably, this mixing does not suppress WAL, suggesting contrasting interference physics compared to graphene. We identify scattering among axially spin-momentum locked states as a key process that leads to a spin-orbital entanglement.

[1] Max Planck Institute for Solid State Research, 70569 Stuttgart, Germany. [2] Department of Physics, Harvard University, Cambridge, MA 02138, USA. [3] L. D. Landau Institute for Theoretical Physics RAS, 119334 Moscow, Russia. [4] Institute of Physics, Bhubaneswar 751005, India. [5] Homi Bhabha National Institute, Mumbai 400085, India. [6] Department of Physics, University of Tokyo, 113-0033 Tokyo, Japan. [7] Institute for Functional Matter and Quantum Technologies, University of Stuttgart, 70569 Stuttgart, Germany. [8] Present address: Department of Physics, University of Arkansas, Fayetteville, AR 72701, USA.
✉email: hnakamur@uark.edu

Electronic systems with Dirac or Weyl dispersion are characterized by pseudospin degrees of freedom[1–3], whose existence can be detected by techniques sensitive to the phase of the electron wavefunction. In magnetotransport measurements, the phase can be probed via the quantum interference of electron waves that occurs between electrons traveling the same closed path in opposite directions. For normal electrons with weak spin–orbit coupling, the interference causes weak localization (WL). The presence of a Berry curvature in momentum space may lead to an extra phase shift of π for such closed trajectories, resulting in weak antilocalization (WAL), as demonstrated for graphene[4–8]. This phase shift is a direct consequence of pseudospin-momentum locking. Magnetic field breaks time-reversal symmetry required for the interference, thus providing a sensitive probe for the quantum interference: a positive (negative) magnetoconductance follows as a result of WL (WAL).

The role of the valley degrees of freedom in quantum interference effects has been extensively studied in graphene[4–6], and also recently, in other systems including Weyl semimetals[9,10] and transition metal dichalcogenides[11,12]. For graphene, scattering between different valleys scrambles the pseudospin information and causes the system to revert back to WL (Fig. 1a). More specifically, intervalley scattering causes a crossover from the symplectic time-reversal symmetry, which characterizes the emergent degrees of freedom in individual valleys (pseudospin), to orthogonal time-reversal symmetry, which characterizes the microscopic degrees of freedom (real spin). The underlying reason for this phenomenon is that the real spin does not play a significant role in graphene due to negligible spin–orbit interactions; the spin retains full rotational symmetry.

In contrast to graphene, Dirac materials with heavy elements may possess strong spin–orbit coupling (SOC) that could lead to broken spin symmetry. As a result, antilocalization in a Dirac semimetal may persist in the presence of intervalley scattering. Thus, one cannot deduce the degrees of freedom responsible for quantum interference from the sign of the magnetoconductance alone. Instead, WAL can be attributed to one of the two distinct scenarios: (i) pseudospin-momentum locking within isolated Weyl–Dirac nodes or (ii) real spin-momentum locking due to SOC. Despite the recent experimental observation of antilocalization in magnetotransport measurements for Weyl/Dirac semimetals[13–19], the distinction between the aforementioned scenarios, which entails an understanding of the role of the valley degrees of freedom, has not been clarified. Elucidating the origin of the observed antilocalization, and the role of pseudospin and valley degrees of freedom is the main objective of this work.

The band degeneracy induced by the existence of inversion (P) and time-reversal (T) symmetries in the absence of spin-rotation symmetry makes spin–orbit coupled Dirac materials unique in comparison with both graphene, where spin is conserved, and Weyl semimetals such as TaAs[20–23], where the band degeneracy is lifted due to the broken T or P symmetries. Unlike in graphene, a global momentum-independent spin operator commuting with the Hamiltonian cannot be defined for a Dirac material with strong SOC. As a result, the electron spin is entangled to its momentum in a sense that will be explained in detail in this work. Such a "hidden" spin-momentum entanglement is challenging to observe in spin-resolved ARPES measurements[24] due to the existence of both spin states at every momentum. The situation is very different in Weyl semimetals in which a spin texture could be readily measured due to lifted degeneracy. Quantum interference measurements are ideally suited to detect such hidden entanglement in Dirac materials, since WAL is expected whenever spin symmetry is broken, regardless of the existence of PT symmetry.

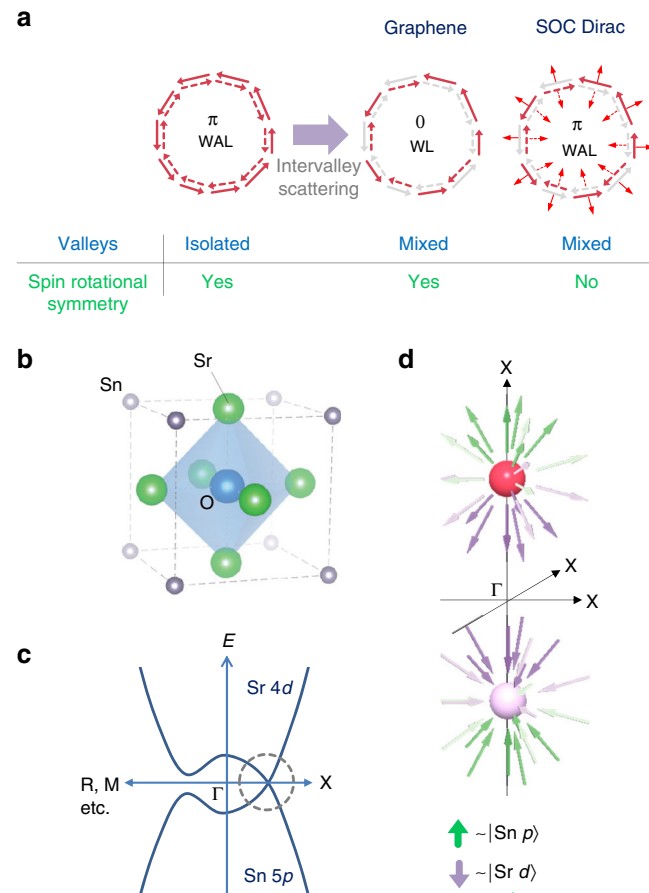

**Fig. 1 Pseudospin and spin in antiperovskites. a** Pseudospin-momentum locking induces a π phase shift for the two electron trajectories (antilocalization). Intervalley mixing causes weak localization for a system with spin-rotational symmetry. A Dirac system with strong spin–orbit coupling could give rise to antilocalization when valleys are mixed since spin-rotational symmetry is broken. **b** Antiperovskite structure of Sr₃SnO. The anion (O) is surrounded by cations (Sr), unlike in normal perovskites, where anion and cation positions are exchanged. **c** Highly schematic diagram showing band inversion of Sr and Sn bands, which produces Dirac nodes along Γ–X momentum directions. **d** Two of the six Dirac pockets which form under moderate hole doping are shown. The arrows represent pseudospins consisting of wavefunctions originating mainly from Sn p states and Sr d states.

Here, we perform a systematic study of the quantum interference effects in a 3D SOC Dirac material Sr₃SnO as a function of doping, which tunes the Fermi energy measured from one of the Dirac nodes ($E_F = -30$ to $-180$ meV). We find dominant negative magnetoconductance (MC) from antilocalization for films at all hole dopings. The magnetoconductance for the lowest $|E_F|$ could be fit to a theoretical model assuming a single interference channel, suggesting that the multiple Dirac valleys are mixed. The observed WAL is thus ascribed to the coupling of real spin, rather than pseudospin, to momentum. Scattering among states with orthogonal spin-quantization axes are responsible for WAL. This scenario describes the robustness of WAL in the whole $E_F$ range studied experimentally.

## Results

**Band structure and the properties of Sr₃SnO thin films.** Sr₃SnO is a member of the family of 3D Dirac materials with antiperovskite

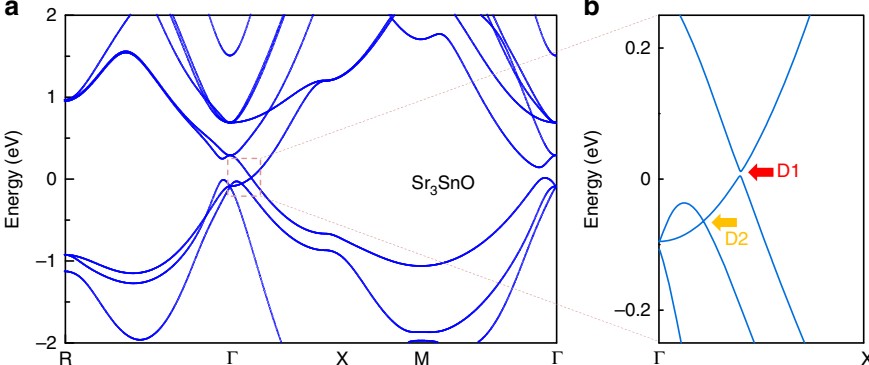

**Fig. 2 Antiperovskite electronic band structures. a** Band diagram of $Sr_3SnO$ obtained from first-principles calculations. **b** The enlarged band diagrams near Dirac nodes. The large arrows indicate the location of Dirac nodes.

structure[25–36]. In this material, six Dirac nodes located along the equivalent $\Gamma$–$X$ directions (due to cubic symmetry) form via the band inversion of the Sr $4d$ and Sn $5p$ bands (Fig. 1c). For a lightly hole-doped case, this results in the presence of six Fermi pockets. The chirality for each valley can be defined using two bases composed mainly of Sn $5p$ and Sr $4d$ states for $Sr_3SnO$, as shown schematically in Fig. 1d.

The first-principles calculations for $Sr_3SnO$ are shown in Fig. 2a, b. The result indicates two sets of Dirac electrons for $Sr_3SnO$: one with a small gap (D1) common to other antiperovskites[25–28], and the other (D2) without a mass gap (Fig. 2b). The second Dirac points (D2) is found to be strictly protected by symmetry even after the inclusion of SOC, which was also discovered in a related material $Ba_3SnO$[34]. Both of these Dirac nodes (D1 and D2) are located along the $\Gamma$–$X$ lines in the Brillouin zone with six copies of each due to cubic symmetry.

$Sr_3SnO$ films were grown by a molecular beam epitaxy[30]. The growth was performed on YSZ (yttria stablized $ZrO_2$) substrates at a substrate temperature of 450 °C. The grown films were single-crystalline with no impurity phase in the X-ray diffraction (Supplementary Figure 1). We grew and analyzed in total 13 films, with thicknesses $d = 50$–$300$ nm. The carrier densities and mobilities obtained from the Hall effect at 10 K were 1.7 $-13 \times 10^{19}$ cm$^{-3}$ and 18–270 cm$^2$/Vs. The sign of the Hall effect was always positive, indicating that all the films were hole doped. By adjusting the Sr/Sn ratio during the growth, the hole density $n_p$ was controlled, which allowed systematic tuning of $E_F$ ($-30$ to $-180$ meV). These $E_F$ values were obtained by integrating the density of states from first-principles calculation until it matched experimental hole densities $n_p$.

**Magnetoconductance and dimensionality of WAL**. Figure 3 shows differential MR for two representative samples with different thicknesses (105 and 50 nm). A magnetic field was applied perpendicular ($\theta = 90°$) or parallel to the current ($\theta = 0°$). We observed a clear positive MR in the low-field region for both thick and thin films, which we attribute to weak antilocalization (Fig. 3a, b). Notably, for the 105-nm-thick film, the low-field MR ($B < 0.5$ T) was insensitive to the relative angle between the magnetic field and the current direction (Fig. 3c), which suggests that the WAL is three dimensional in origin. In contrast, for the thinner film ($d = 50$ nm), although wide-field MR is very similar for the two orientations (Fig. 3b), the in-plane WAL has a broader lineshape than the out-of-plane WAL in a low-field magnified plot (Fig. 3d). This makes sense, because we expect a flat line in the pure 2D limit[37]. Thus, we interpret that the films with thickness $d \sim 100$ nm or greater are in a 3D regime for localization, whereas those with $d \sim 50$ nm are in a quasi-2D regime. We

note that 50 nm is currently the thinnest film we can achieve without degradation.

The magnetoconductance, $\Delta\sigma$, taken for $Sr_3SnO$ films with different carrier densities is shown in Fig. 4a–d. In these plots, $\Delta\sigma$ measured in units of $e^2/\pi^2 h$ μm$^{-1}$, where $h$ is the Planck constant and $e$ is unit charge. WAL appears as a sharp peak around zero field at low temperatures, and is seen for the whole range of carrier density. A negative magnetoconductance proportional to $B^2$ is observed at higher fields. This contribution dominates the entire field range at higher temperatures ($T > 100$ K), and originates from classical orbital effects due to the Lorentz force. In addition, $\Delta\sigma$ develops clear positive slopes in intermediate field region for larger $n$, which is especially pronounced at the highest doping levels (Fig. 4c, d). We attribute this to a crossover to WL from the low-field WAL.

**Quantum interference channels**. The following equation describes the field dependence of $\Delta\sigma$:

$$\Delta\sigma(B) = N\frac{e^2}{4\pi h l_B}\left[2\zeta\left(\frac{1}{2}, \frac{1}{2} + \frac{l_B^2}{l^2}\right) + \zeta\left(\frac{1}{2}, \frac{1}{2} + \frac{l_B^2}{l_\phi^2}\right) - 3\zeta\left(\frac{1}{2}, \frac{1}{2} + 4\frac{l_B^2}{l_{SO}^2} + \frac{l_B^2}{l_\phi^2}\right)\right]. \quad (1)$$

Here, $N$ is the number of independent interference channels, $\zeta$ is the Hurwitz zeta function, and $B$ is the magnetic field. The length scales $l_B = \sqrt{\hbar/4eB}$, $l$, $l_\phi$, and $l_{SO}$ denote the magnetic length, the mean free path, the phase coherence length, and the spin–orbit scattering length, respectively. Equation (1) is an extension of the Hikami–Larkin–Nagaoka theory[38] of weak (anti)localization to 3D, and the spin–orbit-free WL formula ($l_{SO}^{-1} = 0$) derived by Kawabata[39,40]. We also note that the strong SOC limit ($l_{SO}^{-1} = \infty$) of Eq. (1) coincides with a formula used to describe WAL in Weyl semimetals[9]. The prefactor $N$ corresponds to a valley degeneracy factor in the limit of negligible intervalley scattering[41–43]. The derivation of Eq. (1) is provided in Supplementary Note 1.

To analyze the experimental MC, we consider the lowest magnetic fields when $l_B \gg l$, $l_{SO}$. As we discuss in detail in the Supplementary Note 1, the second zeta function in Eq. (1) becomes the dominant term:

$$\Delta\sigma(B) \sim N\frac{e^2}{4\pi h l_B}\zeta\left(\frac{1}{2}, \frac{1}{2} + \frac{l_B^2}{l_\phi^2}\right). \quad (2)$$

Hence, the low-field data are fitted by a function with only two independent parameters: number of channels $N$, and dephasing length $l_\phi$. The results of the fit using Eq. (2) are shown in

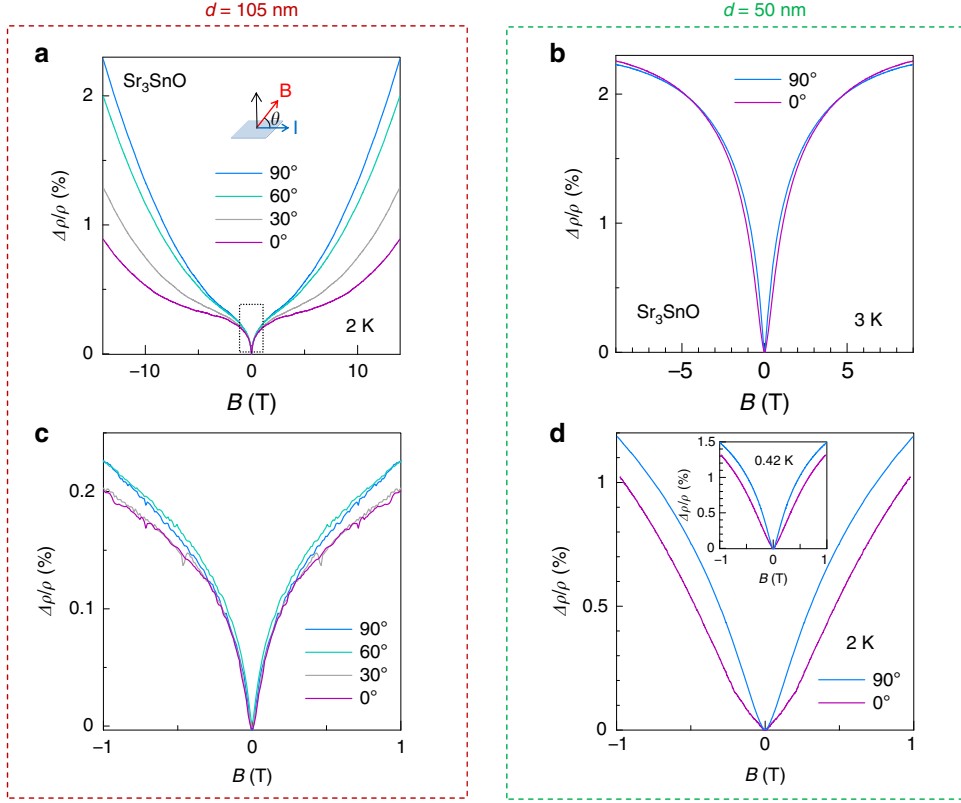

**Fig. 3 Angle dependent MR of Sr$_3$SnO taken at low temperatures.** The MR is normalized with resistance at zero magnetic field. **a** MR for 105-nm-thick film and **b** 50-nm-thick film for a wide-field range ($B > 9$ T). The inset in (**a**) shows the orientation of current (**I**) and magnetic field (**B**) with respect to thin film (shown as a slab). **c** Zoom-in of the low-field MR indicated with dashed square in (**a**). **d** The low-field MR for the 50-nm-thick film at 2 K. The inset of (**d**) shows MR at 0.42 K, in which the qualitative difference between the data for different angles remain unchanged compared with that of 2 K.

Fig. 4e–o. From the estimate of the mean free path for each sample ($l = 0.9$–4.4 nm), the low-field regime used for the fit was chosen to be $B < 0.1$–0.5 T ($l_B = 41$–18 nm), corresponding to $l_B$ approximately a factor of 10 larger than $l$. For samples with $N \sim 1$ (low $n_p$), the absence of WAL–WL crossover enables us to establish an upper bound on the value of $l_{SO}$ which is shorter than the magnetic length $l_B$, thus the condition $l_B \gg l_{SO}$ is manifestly fulfilled. For higher $n_p$ data, we estimate $l_{SO} < 35$ nm by using a full formula (Eq. (1)). Although this is comparable to $l_B$ at the largest $B$ used for each fitting, data points at lower $B$ (i.e., larger $l_B$) are more important for WAL/WL due to their steep dependence on $B$, making the fitting based on Eq. (2) robust. Indeed, the fit based on Eq. (2) reproduces the low-field part of the experimental MC data perfectly (Fig. 4e–o), underscoring that the signal originates from WAL as described in Eq. (2). Further details of the fitting procedure and error estimate are described in Supplementary Note 2. The number of quantum channels extracted from the analysis evolves with carrier density (Fig. 4r). At low $n_p$, $N$ takes values close to 1. At higher $n_p$, however, larger $N$ values of between 2 and 3 are deduced. For quasi-2D films, 2D WL/WAL analysis using a HLN formula[38] also gave $N$ approaching 1 at low $n_p$ (Fig. 4p). For completeness, the analysis of quasi-2D films based on the 3D formula is given in the Supplementary Note 3. Fittings results on each film using 3D and 2D formula are summarized in the Supplementary Tables 1 (3D films) & 2 (quasi-2D films).

The evolution of the Fermi surface as a function of $E_F$ derived from the first-principles calculation is shown in Fig. 5. The $E_F$ is measured from the top of the valence band. This corresponds to the energy distance from the first Dirac nodes (D1). At very low $E_F$ ($-5$ meV), D1 forms separate Dirac valleys as expected. With small doping ($E_F = -25$ meV), which approximately corresponds to the lowest $n$ in our experiment, another set of trivial electron pockets, which are separated from D1, emerge. Further doping merges these pockets together and forms a large Fermi sphere centered at the $\Gamma$ point, while small D2 pockets appear. For $E_F = -180$ meV, which corresponds to the highest $n$ studied in this work, two large Fermi surfaces and a small Fermi surface are found, all centered at $\Gamma$ point.

By comparing the evolutions of $N$ and band structure with hole doping, we arrive at the following dichotomy: At low $n_p$, $N \sim 1$. This implies that the six Dirac valleys and the trivial electron pockets are all mixed by scattering. Beyond a critical doping of $n_c = 3$–$4 \times 10^{19}$ cm$^{-3}$, however, $N$ clusters between 2 and 3 (average of 2.4). This implies that the new Fermi pockets which emerge beyond $n_c$ do not fully mix with the preexisting pockets in the presence of disorder, but instead contribute to additional quantum interference channels. Generally, intraorbital scattering dominates interorbital scattering[44,45], and if the new pockets have different orbital characters, then scattering with the preexisting pockets may be suppressed. The non-integer value of $N \sim 2.4$ may be explained as follows: (1) There is small but non-negligible interorbital scattering, such that the new pockets do not form a fully independent interference channel, and/or (2) the different channels have different sets of parameters, requiring a more complicated fit. From here on, we focus solely on the physics in the low $n_p$ regime, where $N \sim 1$.

**Phase coherence lengths**. The phase coherence lengths $l_\phi$ extracted from the fits are plotted in Fig. 6 for films with two different thicknesses ($d = 200$ and 50 nm). Both films had a low

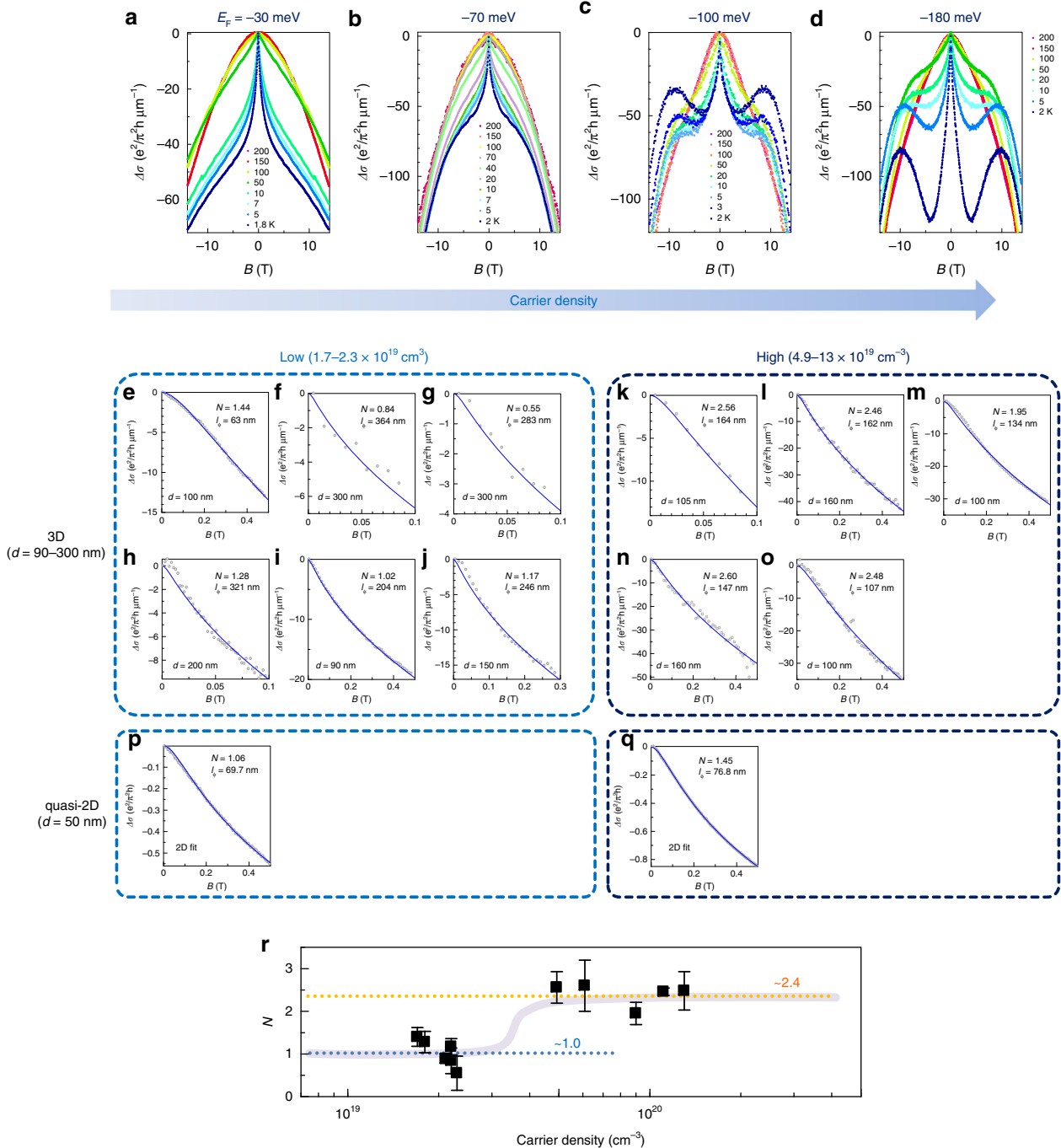

**Fig. 4 Carrier density evolution of the magnetoconductance in Sr₃SnO thin films. a–d** Wide-field ($|B| < 14$ T) MC for four samples with different carrier densities. The Fermi energy estimated from the carrier density (see text) is shown for each graph. Magnetic field orientations are **a** $\mathbf{B} \perp \mathbf{I}$ and **b–d** $\mathbf{B} \parallel \mathbf{I}$. Low-field MC for 3D films in the **e–j** low carrier density regime and **k–q** high carrier density regime. Experimental data are shown in open circles, whereas theoretical fits are shown in solid lines. Each panel represents a different sample. **p, q** Low-field MC for quasi-2D films ($\mathbf{B} \perp \mathbf{I}$), in which the theoretical fits are performed based on a 2D (HLN) formula. **r** The number of channels ($N$) as a function of carrier density. The average value of $N$ for the two different $N$ regimes are shown accompanied with dotted lines. The gray line is a guide to the eye. The error bars represent standard deviation. Source Data are provided as a Source Data file.

Hall carrier density of $n_p = 1.8 \times 10^{19}$ cm$^{-3}$. The temperature dependence of $l_\phi$ in general reflects the dephasing due to inelastic scattering. Theoretically, it follows $l_\phi \propto T^{-p/2}$ ($\tau_\phi \propto T^{-p}$), where $p$ depends on the dephasing mechanism[46,47]. In three dimensions, $p = 3/2$ for electron–electron interactions and $p = 3$ for electron–phonon interactions[46,47]. However, note that the latter is for a clean limit, and for a disordered system, predicted values of $p$ ranges from 2 to 4[48]. In 2D, dephasing at low temperatures is

dominated by electron–electron interactions and $p = 1$. From Fig. 6, it can be seen that $l_\phi$ increases with decreasing $T$, with exponents $-0.75$ ($p = 3/2$) and $-0.5$ ($p = 1$) for 3D and quasi-2D films, respectively. This suggests that electron–electron interactions could be a dominant dephasing mechanism in both cases, although we note that dephasing due to electron–phonon interactions in a disordered 3D system is not well understood[48]. Below 5 K, there is a change in exponent, followed by a possible

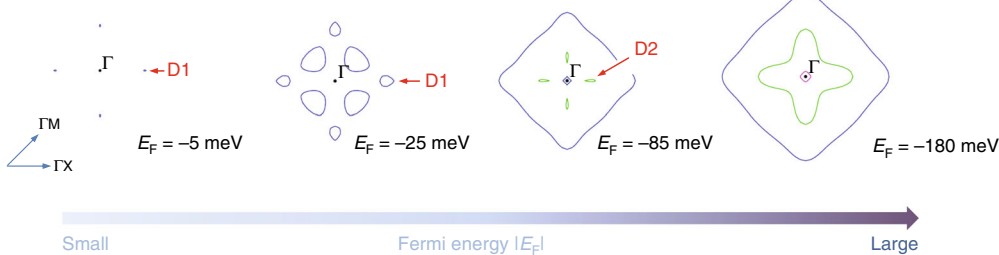

**Fig. 5 Evolution of Fermi surface in Sr₃SnO with $E_F$ obtained by first-principles calculation.** Two different Dirac nodes D1 and D2 appear successively in Sr₃SnO.

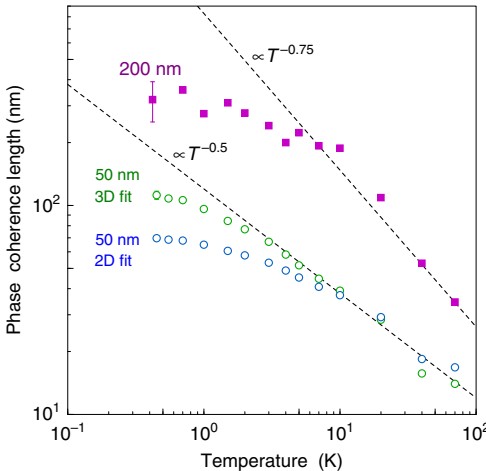

**Fig. 6 Temperature dependence of phase coherence length in Sr₃SnO obtained from localization analysis.** Filled symbols represent data for a 200-nm-thick film analyzed by the 3D localization formula, whereas open symbols represent those for a 50-nm-thick film analyzed by the 3D formula and 2D formula (HLN formula). Error bars are estimated at the lowest temperature for each data using standard deviation. For the 50-nm film, the error was comparable or smaller than the size of the open symbols. Theory predicts dephasing from electron–electron interaction to follow $l_\phi = T^{-0.75}$ in 3D systems (upper dashed line), while $l_\phi = T^{-0.5}$ in 2D systems (lower dashed line). Source Data are provided as a Source Data file.

saturation of $l_\phi$ around 0.5 K. Regarding a possible saturation in $l_\phi$ at low temperatures, in general, it could be attributed to a dephasing which is constant in $T$. Such a source of dephasing includes magnetic spin-spin scattering process[49–52], but many other mechanisms have also been discussed[48].

## Discussion

The experimental results demonstrate extremely robust antilocalization. It not only survives strong intervalley scattering inferred from the valley parameter $N = 1$, but it is also observed at high carrier densities where the different pockets are merged and the non-Dirac bands dominate. The role of real spin in antilocalization needs to be considered to describe this observation.

To this end, we refer to the microscopic wavefunctions realized in Sr₃SnO, whose character is depicted in Fig. 7a. Here, the $z$ axis points toward one of the $X$ directions in the Brillouin zone. We assume a lightly hole-doped situation in which small Dirac pockets form. In this setup, the $p_x \pm i p_y$ states originating from Sn $5p$ orbitals forms the "north pole" of the Dirac pocket. The "south pole" is formed by Sr $4d$. Because of PT symmetry, all of these states are doubly degenerate. However, we find that the spin direction cannot be free as in the case for graphene.

We first describe the situation for the north pole. The wavefunctions are

$$\left|\tilde{\uparrow}, +\right\rangle = -\frac{1}{\sqrt{2}}(|\text{Sn } p_x \uparrow\rangle + i|\text{Sn } p_y \uparrow\rangle),$$
$$\left|\tilde{\uparrow}, -\right\rangle = \frac{1}{\sqrt{2}}(|\text{Sn } p_x \downarrow\rangle - i|\text{Sn } p_y \downarrow\rangle). \tag{3}$$

Here, $\tilde{\uparrow}$ is the chirality (up), the indices $\pm$ denotes a pair of states related by PT symmetry, and $\uparrow, \downarrow$ is spin. These are $m_j = \pm 3/2$ ($J = 3/2$) states. The key observation is that the quantization axis of orbital angular momentum, $l$, is fixed to the direction of momentum, $k$. This is because the splitting between $m_j = \pm 3/2$ and $m_j = \pm 1/2$ states of Sn $5p$ orbitals at finite $k$ breaks the rotational symmetry of $J = 3/2$. The breaking of the rotational symmetry of $J$, in turn, locks the quantization axis of spin via SOC, thus aligning $s$ parallel to momentum (Fig. 7a). Thus, although the double degeneracy still allows both spin-up and -down states to exist at a $k$ point, there is hidden locking between the spin-quantization axis and momentum in the north pole which is axial in nature. This means that any superposition of the two states in (3) (which represents spin pointing in a generic direction) will have strong spin–orbit entanglement, although each of the two states is a product of a spin part and an orbital part and is thus unentangled.

For the south pole, the situation is drastically different. Here, the wavefunction is a superposition of three $d_{x^2-y^2}$ orbitals centered at three different Sr sites in an antiperovskite unit cell (Sr1, Sr2, and Sr3), whose principal axes are pointing in three orthogonal [100] directions[25]:

$$\left|\tilde{\downarrow}, \pm\right\rangle = \frac{1}{\sqrt{6}}(|\text{Sr1 } d_{y^2-z^2} \uparrow, \downarrow\rangle + |\text{Sr2 } d_{z^2-x^2} \uparrow, \downarrow\rangle$$
$$- 2|\text{Sr3 } d_{x^2-y^2} \uparrow, \downarrow\rangle). \tag{4}$$

Here, $\tilde{\downarrow}$ is the chirality (down), and spin ($\uparrow, \downarrow$) in this case does not have fixed orientation with respect to the orbitals. Note that the dominant amplitude is provided by one of the $d_{x^2-y^2}$ orbitals (Sr3), whose principal axis is parallel to $k_z \parallel [001]$. For simplicity, we use this function to display the south pole state in Fig. 7a. For this state, SOC is weak because this orbital does not form a basis for the orbital angular momentum, $l$. This means that the spin direction is free in the south pole, retaining an approximate spin-rotational symmetry as schematically shown in Fig. 7a. As a result, any superposition of the states (4) is unentangled.

Based on these considerations of the microscopic wavefunctions, we propose that intervalley scattering gives rise to spin–orbit entanglement for the states at the north pole, which is responsible for the WAL observed even when the symmetry related to chirality in individual valleys is lost by valley mixing. The essence of this process is captured by considering a scattering between valleys in orthogonal Γ–X directions. As shown in Fig. 7b, this type of intervalley scattering forces rotation of the

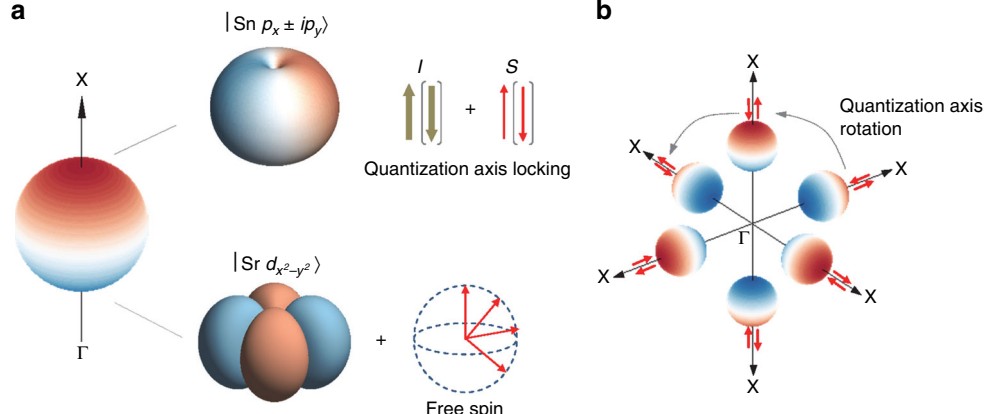

**Fig. 7 Orbital and spin character contributing to the Dirac pockets. a** Magnified view of one of the Dirac pockets. The principal axis of the orbital is locked parallel to momentum for both the north and south poles. In addition, strong spin–orbit interaction locks the spin-quantization axis at the north pole. **b** Intervalley scattering under strong locking of the spin-quantization axis and momentum. Rotation of the spin-quantization axis is expected for scattering between north pole states in Dirac pockets that lie along orthogonal $\Gamma$–$X$ lines (dashed lines). Spin-momentum entanglement follows from such intervalley scattering processes.

spin-quantization axis when states at north poles are involved. As a result, a north pole state for which spin and orbitals are not entangled will become entangled once it is scattered to a north pole state in a different Dirac pocket. It is interesting to note that such spin–orbital entanglement is not introduced when we only consider scattering between a pair of Dirac pockets on the same momentum axis, because the spin-quantization axis remains uniaxial.

The mechanism is readily extended to situations where Dirac pockets are merged. Even in such cases (as shown in Fig. 5), some part of the bands could still have strong $m_j = \pm 3/2$ character from Sn orbitals, especially in the direction along $\Gamma$–$X$. Therefore, scattering which involves such states still could break spin-rotational symmetry and WAL could follow.

Before closing, we make a few remarks regarding possible future extensions of the current work. The crossover between spin-dominated to pseudospin-dominated antilocalization should be observed in the clean limit with negligible intervalley scattering by improving the quality of film. One expects significant enhancement of the antilocalization signal as we approach the clean (pseudospin-dominated) limit due to an increase in the valley factor, $N$. Such crossover may involve a few steps if only a specific pair of valleys (e.g., those along the same $\Gamma$–$X$ line) mix[53]. Evaluation of valley factors in other Dirac-Weyl semimetals with different numbers of valleys will be useful to clarify the role of orbitals in scattering-induced (peudo)spin rotation. Spin-flip impurity scattering was studied in detail for conventional metals and semiconductors[54–59], but little is known about its effects in Dirac-Weyl semimetals. Realistic calculations including the orbital nature of the Dirac-Weyl nodes could thus provide a guideline to utilize such effects in spin transport and other spin-related phenomena.

In conclusion, we have performed a systematic study of quantum interference in a 3D Dirac material with strong SOC across a range of hole carrier concentrations. We observed robust WAL, whose mechanism is distinct from that found in Dirac semimetals with weak SOC (e.g., graphene) or Weyl semimetals with broken $P$ or $T$ symmetries. Our fitting results from the magnetoconductance data reveal that the number of independent interference channels $N$ is greatly reduced by intervalley scattering. This implies that the observed WAL cannot be attributed to the chirality of Dirac electrons, like in graphene, since the pseudospin degree of freedom associated with an isolated valley is quenched. Instead, we propose that a locking of the quantization

axis of the real spin along the axial momentum direction in each Dirac pocket can induce spin–orbital entanglement in the presence of intervalley scattering, thereby restoring WAL. Our measurements thus demonstrate the ability of quantum interference to detect hidden textures in the spin-quantization axis that are invisible to conventional spin-resolved probes. Our work also sheds light on the interplay between real spin and pseudospin in a multivalley Dirac system with strong SOC, which may be manifest in other quantum phenomena, such as spin/pseudospin (Klein) tunneling and the spin/valley Hall effect.

## Methods

**First-principles band calculations**. Self-consistent band structure calculations were performed using the linear muffin-tin orbital (LMTO) method[60] as well as the Wien2k package[61] and consistency of the results has been confirmed. The LMTO method adopted the atomic sphere approximation (ASA) as implemented in the PY LMTO computer code[60]. The Perdew–Wang parameterization[62] was used to construct the exchange-correlation potential in the local density approximation (LDA). Relativistic effects including SOC were taken into account by solving the Dirac equations inside atomic spheres. For the calculation using Wien2k[61], the generalized gradient approximation as parameterized by Perdew–Burke–Ernzerhof[63] was used to describe the exchange-correlation potential. We used atomic sphere radii ($R_{MT}$) of 2.33 (Sr), 2.50 (Sn), 2.33 (O) and $R_{MT}K_{max} = 9.0$, where $K_{max}$ is the plane-wave cut off parameter in the interstitial region outside the atomic spheres. Momentum meshes of $100 \times 100 \times 100$ in the whole Brillouin zone are employed in the self-consistent calculations. Spin–orbit interactions are included using a second-variational method. The orbital character of bands is examined using the result of LMTO code and was in agreement with earlier theoretical works[25,26].

**MBE growth**. Antiperovskite $Sr_3SnO$ was grown by a custom-made molecular beam epitaxy system (Eiko, Japan) at 450 °C[30]. A SrO buffer layer (10 nm) was grown on YSZ substrate at 500–600 °C prior to the deposition of the anti-perovskite film. The elemental flux was controlled to be in the range of 0.015–0.024 Å/s (Sn) and 0.29–0.30 Å/s (Sr) as monitored by quartz crystal microbalance. Diluted oxygen gas (2% in Ar) was introduced using a leak valve. A computer-controlled sequence was used to regulate shutters and oxygen leak valves to separate the oxygen flux from the Sr and Sn fluxes. The main chamber pressure during the introduction of Ar–$O_2$ gas was $1.3 \times 10^{-3}$ Pa, while the background pressure at the deposition temperature was $1$–$2 \times 10^{-6}$ Pa. Films with different carrier densities ($n$) were obtained by adjusting the Sr/Sn flux ratio during the MBE growth. The doping is likely induced by Sr deficiencies: a higher Sr/Sn ratio yielded lower $n$ and the positive sign of the Hall effect (indicating hole as a carrier) is consistent with cation vacancies. Assuming the chemical formula $Sr_{3-x}SnO$ and that one Sr vacancy provides two holes, the observed $n$ corresponds to $x = 0.0016$–$0.0094$.

**Sample preparation and characterization**. After the growth, the film was transferred in vacuum to an Ar glovebox for contact deposition and capping. For XRD, a gold film (80 nm) was deposited uniformly on the film. For transport measurements, Apiezon-N grease was put on the film surface after depositing gold

contacts, but avoiding part of the gold contacts which was later used for electrical connections. The electrical connection was made by wire bonding in air just before installing to PPMS. For some of the films, the connection was made by a silver paste inside the Ar glovebox. The films were characterized by a high resolution four-circle X-ray diffraction system (in-house). Typical XRD $2\theta-\theta$ scans and reflection high-energy electron diffraction (RHEED) image are shown in Supplementary Fig. 1. No impurity peaks are observed in XRD except for those related to the gold capping layer and substrate. Furthermore, RHEED images match the expected in-plane structure of films and indicate an atomically smooth surface. Relatively weak $(00l)$ peaks with odd $l$ for $Sr_3SnO$ reflect a structure factor. From the position of the $(003)$ peaks in XRD, we obtain the lattice constant of the film: $a = 5.13$ Å $(Sr_3SnO)$. The value is very close to those reported for bulk crystals[29]: $a = 5.139$ Å $(Sr_3SnO)$.

## Data availability

The data that support the findings of this study are available from the corresponding author upon reasonable request. The source data underlying Figs. 4r and 6 are provided as a Source Data file.

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

## Acknowledgements

We are grateful to G. Jackeli, T. Kariyado, A. Schnyder, and U. Wedig for helpful discussions. We thank A. Bangura and G. McNally for a critical reading of the paper and discussions. We acknowledge F. Adams, M. Dueller, C. Mühle, and K. Pflaum for technical support. This work was supported by the Alexander von Humboldt-Foundation and the Russian Science Foundation (Grant No. 14-42-00044).

## Author contributions

H.N. designed the research. H.N. and D.H. performed the MBE growth, the transport measurements, and the analysis. J.M. and D.S. contributed in developing antiperovskite MBE. E.K. and P.O. provided the localization formula and theoretical support. A.Y. and H.N. performed first-principles band structure calculations. H.N., E.K., P.O., and H.T. wrote the paper with suggestions from all the other authors.

## Competing interests

The authors declare no competing interests.
