## [Peer Review File · Nature Communications]

Reviewers' comments:

Reviewer #1 (Remarks to the Author):

The authors measured magnetoresistance in a series of samples Sr₃SnO. The measured magnetoresistance varies with the Fermi level by doping, which is evidenced by the measurement of the Hall effect and the first principles band structure calculation. A clear signature of weak antilocalization is revealed in a weak field, independent of the position of the Fermi level and the angle between the magnetic field and electric current. A crossover from the negative magnetoconductivity to a positive magnetoconductivity is observed when the Fermi level shifts away from the Dirac node by doping. The authors regard the positive magnetoconductivity as a signature of weak localization. This work is a combination of experiment, theory and band structure calculation.

1. In Eq. 1, N is the number of independence channel. It is puzzling that, at low carrier density, six Dirac valleys and the trivial electron pocket lead to $N=1$. In the theoretical derivation of the formula in Supplemental Material, it looks that each valley should contribute 1 to N . Due to the double degeneracy of the band structure, the number of the Dirac nodes are always even. It is hard to understand why all Dirac valleys just are mixed to form 1, 2 or 3 channels. This is my main concern on this manuscript. The derivation of the formula in Eq. 1 is based on a single band, NOT multiple bands.

2. The authors ascribed the WAL "to coupling of real spin rather than pseudospin, to momentum". This is a simple an assumption. The band structure calculations do not provide helpful information, and make the problem hard to understand.

3. The authors said "proper values of both N and l_{ϕ} are essential to reproduce the line shape and magnitude of $\Delta \sigma$ ". From the fitting data, all the phase coherence lengths and spin-orbit length for different samples have almost the same order as shown in Fig. 6. However the measured magnetoconductivity has different shape from the low to high density. In Figs. 4e and f, I am curious about the fitting results for a large B , say from 0 to 8T as in Figs. 4g and h. Is there a crossover from WAL to WL? The authors didn't present all the fitting parameters, and explain why there is no crossover from WAL to WL in two samples of lower carrier density.

4. In Fig. 6, what are the fitting exponents for the phase coherent length for the four samples? They should obviously deviate from the theoretical value, -0.75.

The experimental observations look interesting, and the theoretical fitting is very impressive, although the fitting data are not complete. The band structure calculations make the whole picture too vague. I don't think the authors have provided a self-consistent explanation for the measured data.

Reviewer #2 (Remarks to the Author):

In this manuscript, Nakamura et al. reported an electron transport study of 3D Dirac semimetal Sr₃SnO₃. They observed robust negative low field magnetoconductivity for samples with a wide range of chemical potentials. They explained the result with an argument based on intervalley scatterings and spin-orbit coupling. Such a feature distinguishes it from the widely studied graphene, in which the intervalley scatterings leads to a crossover to weak localization (positive magnetoconductivity) due to negligible spin-orbit coupling. The microscopic picture proposed in the manuscript is interesting and plausible. On the experimental side, however, this manuscript needs some improvements or clarifications before I could make a recommendation for publication in Nature Communications. Please see below for details.

(1) The film thickness is in the range of 90-170 nm, which becomes smaller or comparable to the extracted dephasing lengths at liquid helium temperatures. The 3D formula for the weak localization/antilocalization effect (Eq. 1) was used for fitting the data at all temperatures. I strongly suggest similar magnetotransport measurements to be performed on much thinner films (about 20 nm or below), so that the 2D formula can be used to check the appropriateness of the fits. After all, the dephasing length deviates substantially from the $T^{-0.75}$ power law at temperatures below 20 K. The data below 20 K are very important, since the reliability of analyzing the MC data with WL/WAL effect is usually not very good at higher temperatures due to competing effects.

(2) The fits to Eq. 1 required four free parameters, which again call for special care in the data analysis. The authors rely on fixing one of the parameters, channel number N , to integer values. Such an assumption is only valid in the limit of strong inter-channel scatterings. Since the authors argue there is a crossover from $N=1$ to $N=3$ as the Fermi energy is tuned to higher (absolute) values. As shown previously in topological insulators, the inter-channel scatterings can lead to a crossover from $N=1$ to $N=2$ (see e.g. Phys. Rev. B 83, 241304 (2011); Phys. Rev. B 84, 233101 (2011); Phys. Rev. B 86, 035422 (2012)), but it often leads to non-integer values of N if not in the limit of strong or weak inter-channel scatterings. For the magnetoconductivity data shown Fig. 5, how the integer values of N can be justified by just picking four samples with random carrier densities?

(3) It is hard to believe the inter-channel scatterings are so weak in the samples with high hole densities. Some justifications would be helpful.

There are also a few minor points:

(1) In the description of the band structure, the second Dirac point (D2) is mentioned before the first one (D1). It would be better to reverse the order.

(2) More detailed information on how the Fermi energy E_F was evaluated needs to be given.

(3) In the bottom panels of Fig. 5, fitting curves are shown with the raw data only for the lowest temperature. Again, because of the large number of free parameters, it would be better to display fitting curves with the raw data for all temperatures, at least in the Supplementary Information.

Reviewer #3 (Remarks to the Author):

The manuscript by Nakamura et al. reports systematic quantum interference study on anti-perovskite-type 3D Dirac material Sr_3SnO , revealing its hidden entanglement between spin and momentum. By tuning the Fermi energy E_F , they observe evolution of interference effects as a function of E_F , demonstrating robustness of weak antilocalization (WAL) against intervalley scattering, and the contrasting origin of WAL compared to graphene. They suggest spin-momentum entanglement via intervalley scattering induced by an axial spin-momentum locking in each Dirac pocket. I believe this study makes significant progress in the following respects

(1) This study clarifies the origin of WAL in Sr_3SnO and clearly shows that this case is different from that of graphene. At the same time, this study also verifies that Sr_3SnO does not simply follow a scenario of real spin-momentum locking due to spin orbit coupling in a clean limit.

(2) In Sr_3SnO , it is revealed that the intervalley scattering plays an important role in the entanglement between spin and momentum and also occurrence of WAL. This study points out this important aspect for the first time.

(3) All the experiments are well designed and performed, and all the analysis are very solid. Moreover, all the data are quantitatively analyzed based on the complete theory. As a result, physically reliable parameter values were obtained, which becomes basis for reasonable conclusions.

Quantum interference of electron wavefunction in a solid is a longstanding problem. This problem has been renewed since graphene and topological materials were discovered. This is because this interference is significantly influenced by Berry phase and hidden entanglement between spin and momentum. Graphene is a good example that shows the effect of π Berry phase. The present manuscript demonstrates not only this entanglement, but also interplay between this entanglement and disorder. Because of this and the reasons mentioned above, I recommend the publication of the present manuscript in Nat. Commun without any revision.

Below, we provide a point-by-point reply to the concerns raised by the reviewers.

Reviewer #1 (Remarks to the Author):

Reviewer's comment: *1. In Eq. 1, N is the number of independence channel. It is puzzling that, at low carrier density, six Dirac valleys and the trivial electron pocket lead to $N=1$. In the theoretical derivation of the formula in Supplemental Material, it looks that each valley should contribute 1 to N . Due to the double degeneracy of the band structure, the number of the Dirac nodes are always even. It is hard to understand why all Dirac valleys just are mixed to form 1, 2 or 3 channels. This is my main concern on this manuscript. The derivation of the formula in Eq. 1 is based on a single band, NOT multiple bands.*

Reply: We thank the reviewer for this comment. However, we would like to emphasize that the derivation of Eq. 1 did *not* assume a single band, but instead a single transport channel. The derivation is valid for any set of bands provided that they are sufficiently mixed by disorder. That is, the interband and intraband scattering length are of the same order of magnitude. As a result, we generally do not expect the parameter N to correspond to the number of the bands in the system. Instead, it provides the number of transport channels which are not mixed by disorder in the system.

We would also like to point out that, although for intermediate and large fields, Eq. 1 has four independent fitting parameters, at small fields it only depends on N and l_ϕ . Furthermore, its dependence on l_ϕ is very weak (since it mainly acts as a cut off for very small fields). Thus, for reasonable values of l_ϕ , N is the only fitting parameter for the data at small fields. The exceptional quality of the fit at small fields, particularly at low carrier density (for sample #1), provides strong evidence that $N=1$ in this case, implying that all the bands are strongly mixed. **A paragraph was added to the main text to clarify this point (page3, highlighted in blue). In addition, we extended the Supplementary Material to include the discussion on the low-field limit (section B, after Eq. 15).**

It should be noted that having strong mixing between the valleys is not unusual and it has been observed in graphene [e.g. Phys. Rev. Lett. 100, 056802 (2008); PRL 103, 226801 (2009)].

Regarding this issue, we also point out the possibility that we maybe in a regime of Dirac pockets without trivial electron ones for low carrier density. This makes the interpretation of $N=1$ more straightforward. True band structure of Sr_3SnO is yet to be clarified experimentally, and there is always some uncertainty in the first principles calculation: Depending on whether we use GGA or LDA, Sn p and Sr d bands shift slightly relative to each other. The use of more sophisticated exchange-correlation functional, e.g. a hybrid functional, can shift Sn p states even stronger. Therefore, for the low carrier density regime where we deal with a topology change within the energy scale of tens of meV, true Fermi topology could be different. **A remark on this point was added to the main text (page4, left column; highlighted in blue).**

Reviewer’s comment: 2. The authors ascribed the WAL “to coupling of real spin rather than pseudospin, to momentum”. This is a simple an assumption. The band structure calculations do not provide helpful information, and make the problem hard to understand.

Reply: Recent theoretical studies on graphene and Weyl semimetals showed that, for the pseudospin to exhibit WAL, valleys should be independent [Refs.4-6,9]. These studies clarified that under frequent inter-valley scattering, the quantum interference coming from pseudospin shows WL. Because our results show robust WAL despite the mixing of valleys (via observation of $N=1$), we cannot attribute WAL to the pseudospin degree of freedom following the theoretical model proposed so far. This is the underlying reason why we ascribed the WAL “to coupling of real spin rather than pseudospin, to momentum”. This is a rather condensed statement to capture the essence, but we also showed detailed microscopic process that could lead to such effect in the main text.

Reviewer’s comment: 3. The authors said “proper values of both N and l_{ϕ} are essential to reproduce the line shape and magnitude of $\Delta \sigma$ ”. From the fitting data, all the phase coherence lengths and spin-orbit length for different samples have almost the same order as shown in Fig. 6. However the measured magnetoconductivity has different shape from the low to high density. In Figs. 4e and f, I am curious about the fitting results for a large B , say from 0 to 8T as in Figs. 4g and h. Is there a crossover from WAL to WL? The authors didn’t present all the fitting parameters, and explain why there is no crossover from WAL to WL in two samples of lower carrier density.

Reply: We first show the result of higher field fitting data for the low density samples (#1 and #2) in Fig. R1. As can be seen, we do not find crossover from WAL to WL in the fitting result at higher field.

Fig. R1 Fitting results for #1 (a) and #2 (b) including high magnetic field. Experimental data are shown in blue circles and the theoretical fits are shown in red solid lines.

As we briefly mentioned in the main text, the existence of a crossover from WAL to WL depended on the balance between the mean free path l and spin-orbit length l_{SO} . To provide criteria on the crossover, we first show the crossover controlled by tuning l_{SO} (Fig.R2).

Fig. R2 **a** $\Delta\sigma$ - B curves for various l_{SO} values under constant l_ϕ (200 nm) and l (1 nm). **b** Magnetic field at local minimum in $\Delta\sigma$ as a function of l_{SO} to display the parameter region showing WAL/WL crossover at finite B .

As expected, the crossover is observed at higher l_{SO} , and below some critical value of l_{SO} , $\Delta\sigma$ show WAL in the whole magnetic field range. Note that near the critical l_{SO} ($\sim 20\text{nm}$), which is close to the value adopted for #1, upturn in $\Delta\sigma$ due to WL is weak. In this condition small $-B^2$ background can suppress such upturn completely. This explains the lack of WAL/WL crossover in #1.

Next, we show a crossover obtained by tuning l at constant l_ϕ and l_{SO} (Fig.R3). We find that the crossover is observed below some critical l ($\sim 10\text{nm}$), close to the value adopted for #2. Again, near the critical value an upturn due to WL is suppressed. This approximately corresponds to the situation in #2.

Fig. R3 **a** $\Delta\sigma$ - B curves for various l values under constant l_ϕ (200 nm) and l_{SO} (30 nm). Magnetic field at local minimum in $\Delta\sigma$ as a function of l to display the parameter region showing WAL/WL crossover at finite B .

Finally, we extend these analyses to show parameter (l and l_{SO}) space required for the observation of crossover. This is performed by fixing other parameters (N , l_ϕ , and C) close to the experimental situations (Fig. R4).

Fig. R4 Parameter (l_{SO} and l) space for the observation of WAL to WL crossover at finite field. Other parameters (N , l_ϕ , and C) are fixed to experimentally relevant conditions for each sample: **a** for #1, **b** for #2 and #3, **c** for #4.

These plots clarify how each sample is separated in a parameter space to allow/forbid the observation of WAL to WL crossover.

The discussion on the crossover behavior was added in the revised Supplementary Material. In addition, we included full parameter information used for fitting in Supplementary Material.

Reviewer's comment: 4. In Fig. 6, what are the fitting exponents for the phase coherent length for the four samples? They should obviously deviate from the theoretical value, -0.75 .

Reply: We address this point in detail below (as a reply to Reviewer #2).

Reviewer #2 (Remarks to the Author):

Reviewer's comment: (1) The film thickness is in the range of 90-170 nm, which becomes smaller or comparable to the extracted dephasing lengths at liquid helium temperatures. The 3D formula for the weak localization/antilocalization effect (Eq. 1) was used for fitting the data at all temperatures. I strongly suggest similar magnetotransport measurements to be performed on much thinner films (about 20 nm or below), so that the 2D formula can be used to check the appropriateness of the fits. After all, the dephasing length deviates substantially from the $T^{-0.75}$ power law at temperatures below 20 K. The data below 20 K are very important, since the reliability of analyzing the MC data with WL/WAL effect is usually not very good at higher temperatures due to competing effects.

Reply: We thank the reviewer for this comment. We first would like to point out experiments which showed deviation from the theoretical exponent at low temperatures as observed in our study (Fig. R5). Although we believe that the understanding of this behavior is not established yet, Engels *et al.* [PRL 113, 126801 (2014)] suggested that the electron spin-flip scattering processes may limit the phase coherence at lower temperature. Although these are for 2D systems, similar physics may contribute to the suppression of phase coherence in 3D.

Regarding the dimensionality issue, the angular dependent study (Fig.3) showed that the MR data for different magnetic field angles perfectly matches at low B (which is the most important region for extracting l_ϕ). If our system is behaving as 2D in terms of localization effects, data taken at different field angles should show difference in curves in the low- B region. We thus associate the saturation behavior at low temperature to an intrinsic property of the electron system, rather than coming from the temperature-dependent crossover of effective dimensions.

[Redacted]

Fig.R5. Phase coherence length/time for (a) monolayer MoS₂, (b,c) bilayer graphene, and (d) Bi₂Se₃. The black solid lines in (b) and (c) correspond to the electron-electron dephasing model developed by Altshuler et al. In all cases, deviation from simple exponent is observed at low temperature ($T \sim < 10$ K). [(a) adopted from PRL 116, 046803 (2016); (b,c) from PRL 113, 126801 (2014); (d) PRB 84, 233101 (2011)].

We certainly agree with the reviewer that checking the 3D to 2D crossover by reducing the film thickness will provide additional insights. We attempted such study in the past, but failed due to extreme reactivity of our films in the thinner limit even after capping. We believe such study should be possible by developing a completely sealed measurement system that enables direct installation to PPMS under vacuum/inert gas. We wish to deliver 3D-2D crossover physics as a separate work in the future.

We added the discussion on the phase coherence at low temperature and a new reference in the revised main text (page5, right column; highlighted in blue).

Reviewer's comment: (2) The fits to Eq. 1 required four free parameters, which again call for special care in the data analysis. The authors rely on fixing one of the parameters, channel number N , to integer values. Such an assumption is only valid in the limit of strong inter-channel scatterings. Since the authors argue there is a crossover from $N=1$ to $N=3$ as the Fermi energy is tuned to higher (absolute) values. As shown previously in topological insulators, the inter-channel scatterings can lead to a crossover from $N=1$ to $N=2$ (see e.g. Phys. Rev. B 83, 241304 (2011); Phys. Rev. B 84, 233101 (2011); Phys. Rev. B 86, 035422 (2012)), but it often leads to non-integer values of N if not in the limit of strong or weak inter-channel scatterings. For the magnetoconductivity data shown Fig. 5, how the integer values of N can be justified by just picking four samples with random carrier densities?

Reply: We thank the reviewer for this useful comment and the reference. We would first like to emphasize that fitting N only requires the data at low fields for which Eq. 1 has only two, rather than four, fitting parameters. Furthermore, the dependence of the result on one of these parameters l_ϕ is rather weak for the physical range of possible values. This means that the assumption that N is an integer is not really needed for the fit. It is instead a feature of the data.

As can be seen from Fig.4e and Fig.R6 below, the value $N=1$ fits the low field data perfectly for the lowest doped sample (#1) where our conclusion is strongest. For samples #2-#4, it is possible that the value of N deviates a bit from the integer values $N=2$ or 3 if we assume non-integer values for N , but again, considering the quality of the fit (Fig.R6), this deviation is expected to be very small (see also our reply to comment #1 from reviewer #1).

Fig. R6 Fitting results for the low field ($B < 0.7$ T) regime. Experimental data (open circles) and theoretical fit (solid lines) are shown for **a** sample #1, **b** sample #2, **c** sample #3, and **d** sample #4.

It is true, as the referee points out, that taking an integer value of N relies on the simplified assumption that the valleys (or Fermi pockets/surfaces) either mixes strongly, or are completely independent. There is no reason a priori to expect this to be the case for our samples, but it is strongly suggested by the data. It should be noted that, although non-integer value of N has been used in the past for 2D systems using an analogous theoretical framework, one major drawback of this approach is that parameters used in a conventional theoretical model (such as l , l_{SO}) lose their physical meaning for non-integer N . This has been explicitly pointed out, for example, in Ref. [Phys. Rev. B 84, 233101 (2011)]. Thus, we stress that although non-integer values of N are used in literature and certainly useful to show transient regime of intermediate scattering strength, we need to be careful about the interpretation of results. Given the quality of our low-field fit using integer N , we saw no need to include such complication by introducing non-integer N .

The arguments on non-integer N , together with new references, are included in the revised main text (page3, right column; highlighted in blue). In addition, the low-field fitting data are included in the revised Supplementary Material.

Reviewer's comment: (3) It is hard to believe the inter-channel scatterings are so weak in the samples with high hole densities. Some justifications would be helpful.

Reply: One possible explanation is that the carriers in the three Fermi surfaces (for higher E_F) experience much frequent *intra*-Fermi-surface scattering compared to *inter*- Fermi-surface scattering. This produces following situation: majority of electron that made a certain closed loop to produce interference did so purely via *intra*-valley scattering, meaning that essentially different Fermi surfaces contributed separately to the quantum interference. In this case, we can attribute $N=3$ to three distinct Fermi surfaces appearing at higher hole doping. Physically, much frequent *intra*-Fermi-surface scattering can be justified if bands with identical orbital character have larger scattering matrix element.

We included this discussion in the revised main text (page4-, bottom right; highlighted in blue).

Reviewer's comment: There are also a few minor points:(1) In the description of the band structure, the second Dirac point (D2) is mentioned before the first one (D1). It would be better to reverse the order.

Reply: We thank the reviewer for this comment. We revised the order as suggested (page2, right column; highlighted in blue).

Reviewer's comment: (2) More detailed information on how the Fermi energy E_F was evaluated needs to be given.

Reply: We included the detailed of the E_F extraction in the revised main text (page2, bottom right; highlighted in blue).

Reviewer's comment: (3) In the bottom panels of Fig. 5, fitting curves are shown with the raw data only for the lowest temperature. Again, because of the large number of free parameters, it would be better to display fitting curves with the raw data for all temperatures, at least in the Supplementary Information.

Reply: Please find below the fitting curves together with raw data for all temperatures (Fig.R7). We added these data in revised Supplementary Material.

Fig.R7. **a-d** Fitting curves superimposed with raw data for all temperatures.

Reviewer #3 (Remarks to the Author):

The manuscript by Nakamura et al. reports systematic quantum interference study on anti-perovskite-type 3D Dirac material Sr₃SnO, revealing its hidden entanglement between spin and momentum. By tuning the Fermi energy E_F , they observe evolution of interference effects as a function of E_F , demonstrating robustness of weak antilocalization (WAL) against intervalley scattering, and the contrasting origin of WAL compared to graphene. They suggest spin-momentum entanglement via intervalley scattering induced by an axial spin-momentum locking in each Dirac pocket. I believe this study makes significant progress in the following respects (1) This study clarifies the origin of WAL in Sr₃SnO and clearly shows that this case is different from that of graphene. At the same time, this study also verifies that Sr₃SnO does not simply follow a scenario of real spin-momentum locking due to spin orbit coupling in a clean limit. (2) In Sr₃SnO, it is revealed that the intervalley scattering plays an important role in the entanglement between spin and momentum and also occurrence of WAL. This study points out

this important aspect for the first time.

(3) All the experiments are well designed and performed, and all the analysis are very solid. Moreover, all the data are quantitatively analyzed based on the complete theory. As a result, physically reliable parameter values were obtained, which becomes basis for reasonable conclusions.

Quantum interference of electron wavefunction in a solid is a longstanding problem. This problem has been renewed since graphene and topological materials were discovered. This is because this interference is significantly influenced by Berry phase and hidden entanglement between spin and momentum. Graphene is a good example that shows the effect of π Berry phase. The present manuscript demonstrates not only this entanglement, but also interplay between this entanglement and disorder. Because of this and the reasons mentioned above, I recommend the publication of the present manuscript in Nat. Commun without any revision.

Reply: We thank the reviewer for pointing out the significance of our work and recommending the publication in Nature Communications.

Reviewers' comments:

Reviewer #2 (Remarks to the Author):

I'm still concerned with the dimensionality of the samples regarding the WAL effect. After all, the dephasing lengths at 2 K extracted for samples #1-#3 are longer or comparable to the film thicknesses, as shown in Fig. 6a. The deviation from the $T^{-0.75}$ law (i.e. signature for Altshuler-Aronov dephasing due to electron-electron interactions) at temperatures below 10 K may not be viewed as a simple saturation behavior due to spin-flipping scattering. The authors pointed out that similar behavior has also been observed in topological insulators (TIs). However, in a recent experiment, the dephasing rate in more carefully controlled TI samples does not show any saturation behavior down to about 0.1 K [Liao et al., Nat. Commun. 8, 16071 (2017)]. Usually the reliability of the dephasing length extracted with the WL/WAL effect is not very good for temperatures above 20 K. I really doubt a well-defined Altshuler-Aronov dephasing can be concluded from the data shown in Fig. 6a. I strongly suggest the authors extend the measurements down to at least 0.3 K, if thinner samples cannot be prepared with satisfactory quality. It is possible that the dephasing rate follows a different power law instead of a crossover to saturation.

I also suggest the authors take caution in the magnetic field range during fitting their data. The WL/WAL effect should be suppressed at fields above certain limits.

Reviewer #4:

Dear Editor,

After reading the manuscript, the first reviews and the present reports I would like to comment on the possibility of a further revision of the manuscript "Robust weak antilocalization due to spin-orbital entanglement in Dirac material Sr₃SnO".

The previous comments of the reviewers have been considered carefully and were reasonably taken into consideration for revising the manuscript. Now, reviewer #1 is concerned about details of the fitting procedure.

In fact, I can confirm that the fit parameters given in section E of the supplementary materials seem questionable and leave serious doubts about the carefulness of this study.

Minor points are:

- The notation in Tables S1-4 differs from the rest of the manuscript (capital letters instead of small letters).
- The values are given in an unphysical notation.

More importantly, the values are not consistent with the description of the fitting procedure in the main text: On page 3 of the main text the authors state that " l was used as a fitting parameter with finite bound (1-10nm)". However, not only is the error of the fit unphysically constant, but also it seems that l was fixed to some other arbitrary value in case of sample #3 (see constant value of $l=7$ in Table S3). These points seem to be minor issues, but they are not. A large part of the authors' conclusions are based on the result of this fitting. For example, the classification of samples #2 and #3 in the "phase diagram" of Fig. S4b is due to the values of 10 and 7 that seem to be fixed by the authors to a guessed value.

Concerning the recent comments of Reviewer #2:

Extending the measurements down to lower temperatures may in fact give additional insights. Due to the fragility of the films, this most likely has to be performed using different samples, which may also yield an extended data basis for the observation of the WAL to WL crossover.

In addition to the other reviewers' comments:

Unfortunately, the authors do not give detailed information about the transport measurements and how the data were obtained. What geometry did the measured samples have? As a consequence: Can the authors fully rule out the effect of "current jetting" [see e.g. R D dos Reis et al 2016 New J. Phys. 18 085006 (2016)] which severely depends on the geometry? Although the effect is expected to be suppressed for 2D systems, the samples discussed here may partially be considered to be 3D.

In summary I think that a revision towards publication is in principle possible, however I would not expect such thoughtless way of data fitting performed by authors that wish to publish in a high-quality journal like Nature Communications.

Reviewer #5's informal comments:

Potentially, the manuscript is interesting, both from a theory perspective (new equations for weak quantum corrections to transport are derived) as well as from the experimental results.

However, when I started reviewing I got stuck immediately. To check whether the observed effects are weak quantum corrections, one wants to see that the conductance changes on the scale of the conductance quantum only. As this is a 3D material, the change in conductivity is given. But the dimensions are incorrect. E^2/h has dimensions Ohm^{-1} already, so the units of $e^2/h \text{ Ohm}^{-1} \text{ m}^{-1}$ make no sense.

Somewhat in line with the other informal comment by reviewer 1, the manuscript cannot be judged on its merits if the basics are not clear. These are probably just due to sloppiness, but as a reader/referee one gets stuck.

Reply to Reviewers

Please find below the reply to each of the referee's comment. The changes made in the main text are highlighted in blue.

Reviewer #2 (Remarks to the Author):

I'm still concerned with the dimensionality of the samples regarding the WAL effect. After all, the dephasing lengths at 2 K extracted for samples #1-#3 are longer or comparable to the film thicknesses, as shown in Fig. 6a. The deviation from the $T^{-0.75}$ law (i.e. signature for Altshuler-Aronov dephasing due to electron-electron interactions) at temperatures below 10 K may not be viewed as a simple saturation behavior due to spin-flipping scattering. The authors pointed out that similar behavior has also been observed in topological insulators (TIs). However, in a recent experiment, the dephasing rate in more carefully controlled TI samples does not show any saturation behavior down to about 0.1 K [Liao et al., Nat. Commun. 8, 16071 (2017)]. Usually the reliability of the dephasing length extracted with the WL/WAL effect is not very good for temperatures above 20 K. I really doubt a well-defined Altshuler-Aronov dephasing can be concluded from the data shown in Fig. 6a. I strongly suggest the authors extend the measurements down to at least 0.3 K, if thinner samples cannot be prepared with satisfactory quality. It is possible that the dephasing rate follows a different power law instead of a crossover to saturation.

(our reply)

We thank the reviewer for detailed comments regarding (i) the dimensionality and (ii) the lack of low temperature data associated with dephasing and saturation effect.

(i) dimensionality

We grew and analyzed two films with thickness of $d=50\text{nm}$, significantly thinner than the thinnest film in the previous version ($d=90\text{nm}$). This was made possible by the revised capping method, where we did all the wiring in the Ar glovebox. By doing so, we observed the effect of finite thickness as shown in Fig.3b,d of the revised main text. For the thinner film ($d = 50 \text{ nm}$), although wide-field MR is very similar for the two magnetic field orientations (Fig. 3b), the in-plane WAL ($B // I$) has a broader lineshape than the out-of-plane WAL in a low-field magnified plot (Fig. 3d). This makes sense, because we expect a flat line in the pure 2D limit. Thus, we interpret that the films with thickness $d = 100 \text{ nm}$ or greater are in a 3D regime for localization, whereas those with $d = 50 \text{ nm}$ are in a quasi-2D regime. We also note that currently 50 nm was the thinnest sample we could grow and measure without degradation. These discussions are added in the revised main text.

Changes made:

[page 2, right column, 2nd line from the bottom]

One paragraph has been added to the main text to describe the MR for a quasi-2D film.

"Fig. 3 shows experimental MR for two representative samples..."

(ii) dephasing

We measured two representative films, with $d=200\text{nm}$ and $d=50\text{nm}$ down to 0.42K and 0.45K , respectively. Note that from the result of angular dependent MR in Fig.3, the 200nm film is well within a 3D WAL regime, whereas the 50nm film is in a quasi-2D regime. The data show that an exponent expected for a dephasing via electron-electron interaction does seem to apply at approximately $5\text{-}50\text{K}$. However, we note that there could be a potential change of exponent at 5K toward low temperatures, followed by possible saturation. In this sense, we cannot exclude the possibility that the phase coherence length follows a different exponent as a function of temperature. All these observations are discussed explicitly in the revised paper, together with added references which will help readers be informed of a variety of interpretations on the dephasing exponent as well as the saturation effect.

Changes made:

[page 2, left column, 8th line from the bottom]

One paragraph has been rewritten to describe new set of data taken down to $\sim 0.4 \text{ K}$ for 3D and quasi-2D films.

"The phase coherence length l_{ϕ} extracted from the fits are plotted in ..."

Reviewer #4:

After reading the manuscript, the first reviews and the present reports I would like to comment on the possibility of a further revision of the manuscript "Robust weak antilocalization due to spin-orbital entanglement in Dirac material Sr_3SnO ".

The previous comments of the reviewers have been considered carefully and were reasonably taken into consideration for revising the manuscript. Now, reviewer #1 is concerned about details of the fitting procedure.

In fact, I can confirm that the fit parameters given in section E of the supplementary materials seem questionable and leave serious doubts about the carefulness of this study.

(our reply)

We thank the referee for these comments. Based on concerns raised regarding our fit parameters, we completely revised the analysis. The new analysis is based on a limiting formula at low field, which was also shown in the previous version. This requires only two fitting parameters, and the same conclusion (establishment of $N=1$ at low n) is derived.

The detail of the fitting procedure, using the low-field formula (Eq. 2), is summarized in the following.

1. Pick a magnetic field range for the fitting for the sample, such that l_b is roughly ten times larger than l (as estimated from transport and band parameters).
2. Fit to determine N and l_ϕ at every temperature.
3. Report N for each sample as an average over a few lowest temperatures, with some standard deviation.
4. Using the average of N determined for the lowest few temperatures, redo the fits and extract l_ϕ versus T for fixed N . (To determine the temperature dependence of l_ϕ , we note that N should physically be a T -independent quantity.)
5. Additional fits were tried up to $\pm 30\%$ of the original magnetic field range to obtain an error due to the range of magnetic field used in the fitting.

In a few cases, using the average value of N caused a larger change in l_ϕ than the error obtained from changing field range, so this was accounted for in the error of l_ϕ .

Thus, the error in N includes considerations from temperature averaging and range of magnetic field used, while the error in l_ϕ includes contributions from errors in N and also the range of magnetic field used.

The details for fitting procedure are added in the revised Supplementary Materials.

Changes made:

[page 3, right column]

New paragraph was added to describe the revised fitting procedure.

"To analyze the experimental MC..."

[Supplemental Materials, Section C]

New section has been added to describe the detail of the fitting procedure and for the estimation of errors.

(Reviewer #4)

Minor points are:

- The notation in Tables S1-4 differs from the rest of the manuscript (capital letters instead of small letters).

(our reply)

In the new analysis, the discussion on I and I_{SO} that were the outcome of a full formula is not relevant and thus omitted.

(Reviewer #4)

- The values are given in an unphysical notation.

(our reply)

We revised the notation for the quantum conductance correction in 3D, both in Fig.4 and in the main text.

(Reviewer #4)

More importantly, the values are not consistent with the description of the fitting procedure in the main text: On page 3 of the main text the authors state that " L was used as a fitting parameter with finite bound (1-10nm)". However, not only is the error of the fit unphysically constant, but also it seems that L was fixed to some other arbitrary value in case of sample #3 (see constant value of $L=7$ in Table S3). These points seem to be minor issues, but they are not. A large part of the authors' conclusions are based on the result of this fitting. For example, the classification of samples #2 and #3 in the "phase diagram" of Fig. S4b is due to the values of 10 and 7 that seem to be fixed by the authors to a guessed value.

(our reply)

In the previous version, an error was estimated by performing a fit using four different magnetic field ranges. These were performed fully automatically using a least-squares code. The problem was that L showed in general very large variation, and when combined with the upper bound of 10nm, it seems to have caused an artificial mean value that returns, e.g. $L=7$. Because of the large number of fitting parameters (five in older analysis) and the complexity involved in analyzing the error in such situations, we decided to adopt only the simple limiting formula with two variables in the revised manuscript, N and I_{ϕ} , which suffice to discuss the key physics. In addition, a detailed description of how we derived the error bars has been added to Supplementary Materials.

Changes made:

[page 3, right column]

New paragraph was added to describe the revised fitting procedure.

"To analyze the experimental MC..."

[Supplemental Materials, Section C]

New section has been added to describe the detail of the fitting procedure and for the estimation of errors.

(Reviewer #4)

Concerning the recent comments of Reviewer #2:

Extending the measurements down to lower temperatures may in fact give additional insights. Due to the fragility of the films, this most likely has to be performed using different samples, which may also yield an extended data basis for the observation of the WAL to WL crossover.

(our reply)

We thank the referee for this suggestion. We have extended our measurement down to lower temperatures ($T \sim 0.4\text{K}$). We do note, however, that our main focus regarding this work is not the WAL to WL crossover but the robust WAL under the mixing of multiple valleys as evidenced by our analysis on quantum channels.

(Reviewer #4)

In addition to the other reviewers' comments:

Unfortunately, the authors do not give detailed information about the transport measurements and how the data were obtained. What geometry did the measured samples have? As a consequence: Can the authors fully rule out the effect of "current jetting" [see e.g. R D dos Reis et al 2016 New J. Phys. 18 085006 (2016)] which severely depends on the geometry? Although the effect is expected to be suppressed for 2D systems, the samples discussed here may partially be considered to be 3D.

(our reply)

We have performed all the measurements using contacts patterned in a Hall bar geometry (6-pads) by using lithographically defined stencil shadow masks. We have previously performed comparative measurement using a van der Pauw geometry, and in that case we did observe a current jetting effect as we expected: Magnetoconductance followed $\Delta\sigma = +B^2$ at higher magnetic fields (opposite in sign compared to MC coming from Lorentz force). We note that this trend is not observed in our samples shown here, and we exclude the current jetting effect in samples presented in this paper.

In summary I think that a revision towards publication is in principle possible, however I would not expect such thoughtless way of data fitting performed by authors that wish to publish in a high-quality journal like Nature Communications.

Reviewer #5's informal comments:

Potentially, the manuscript is interesting, both from a theory perspective (new equations for weak quantum corrections to transport are derived) as well as from the experimental results.

However, when I started reviewing I got stuck immediately. To check whether the observed effects are weak quantum corrections, one wants to see that the conductance changes on the scale of the conductance quantum only. As this is a 3D material, the change in conductivity is given. But the dimensions are incorrect. E^2/h has dimensions Ohm^{-1} already, so the units of $e^2/h \text{ Ohm}^{-1} \text{ m}^{-1}$ make no sense.

(our reply)

We thank the referee for pointing out this error. The referee is correct. We revised the label on Fig. 4 as well as the unit show in the main text.

Somewhat in line with the other informal comment by reviewer 1, the manuscript cannot be judged on its merits if the basics are not clear. These are probably just due to sloppiness, but as a reader/referee one gets stuck.

Reviewers' comments:

Reviewer #2 (Remarks to the Author):

The revised manuscript is much improved over the previous one. Even though the thicknesses of the films studied in this work are not perfect for rigorous WAL fittings in either 3D or 2D regime, the authors have carried out the fittings in both regimes and the conclusion seems to be plausible. The strength of this manuscript is that the hidden role of spin-momentum locking is unveiled for topological semimetals. Therefore, I recommend publication of this manuscript in Nature Communications,

I only have the following two minor points:

1. For the MC data presented in Fig. 4, it will be helpful to readers if the thicknesses of all films can be given. In addition, it will also be useful if the results of both 3D and 2D fittings to the WAL formulae can be given for each of the samples, preferably in a table of the Supplemental Material.

2. Page 1, column 1, paragraph 2, line 5: there is a typo: "Sn 5d bands".

Reviewer #4 (Remarks to the Author):

The authors have met the previous concerns about the fitting procedure by using the low-field limit and clarifying the details of the data analysis. These changes have strongly increased the transparency of the analysis. Also, the suggestions to investigate thinner films and to widen the temperature range have been followed. The main conclusion of the paper has been unaffected by these changes, yet it is easier for the reader to follow now. My only concern is the following: The authors admit that Eqn. 2 is at the border of its applicability, because I_{SO} is close to I_B , but nevertheless use it for fitting as it "reproduces the low-field part of the [...] data". Is there a more physical justification for its usage? Would the value of N be robust when employing the next order? If the authors could convincingly clarify this detail, I can recommend publishing the manuscript.

Reply to Reviewers

We are grateful to the reviewers for evaluating the revised manuscript. We reply to the remaining suggestions from the referees below.

Reply to Reviewer #2

Reviewer #2 (Remarks to the Author):

The revised manuscript is much improved over the previous one. Even though the thicknesses of the films studied in this work are not perfect for rigorous WAL fittings in either 3D or 2D regime, the authors have carried out the fittings in both regimes and the conclusion seems to be plausible. The strength of this manuscript is that the hidden role of spin-momentum locking is unveiled for topological semimetals. Therefore, I recommend publication of this manuscript in Nature Communications,

We thank the referee for pointing out the scientific essence of our work, and the recommendation for publication.

I only have the following two minor points:

1. For the MC data presented in Fig. 4, it will be helpful to readers if the thicknesses of all films can be given. In addition, it will also be useful if the results of both 3D and 2D fittings to the WAL formulae can be given for each of the samples, preferably in a table of the Supplemental Material.

We inserted the thicknesses for all the films in Fig. 4. We also inserted a table in the Supplemental Material to show results of both 3D and 2D fittings for each sample, following the reviewer's suggestion.

2. Page 1, column 1, paragraph 2, line 5: there is a typo: "Sn 5d bands".

We thank the referee for locating this error. We revised the typo.

Reply to Reviewer #4

Reviewer #4 (Remarks to the Author):

The authors have met the previous concerns about the fitting procedure by using the low-field limit and clarifying the details of the data analysis. These changes have strongly increased the transparency of the analysis. Also, the suggestions to investigate thinner films and to widen the temperature range have been followed. The main conclusion of the paper has been unaffected by these changes, yet it is easier for the reader to follow now.

We thank the reviewer for the positive comments regarding our revision.

My only concern is the following: The authors admit that Eqn. 2 is at the border of its applicability, because I_{SO} is close to I_B , but nevertheless use it for fitting as it "reproduces the low-field part of the

[..] data". Is there a more physical justification for its usage? Would the value of N be robust when employing the next order? If the authors could convincingly clarify this detail, I can recommend publishing the manuscript.

We thank the reviewer for this comment. There are indeed physical reasons behind the robustness of the fitting. Namely, although the criteria ($l_B \gg l_{SO}$) is marginally fulfilled for the data points at the maximum magnetic fields (B), for most of the data points at lower B , $l_B \gg l_{SO}$. Furthermore, the data points at lowest B are the ones that have largest influence on the fitting.

The above discussion applies to $N > 1$, where the crossover to WL is observed, from which we could estimate l_{SO} . For samples with $N \sim 1$ (low n_p), the absence of WAL-WL crossover enables us to establish an upper bound on the value of l_{SO} which is shorter than the magnetic length l_B , thus the condition $l_B \gg l_{SO}$ is manifestly fulfilled.

We also would like to point out that representative fittings based on full formula (Eq. 1), free from the criteria ($l_B \gg l_{SO}$), gave N values consistent with the low- B analysis (Eq.2), although the former was limited to integer N due to the complexity of fitting. Thus, the value of N does not depend on the detail of the analysis (Eq. 1 or 2).

The text in our earlier draft on the criteria ($l_B \gg l_{SO}$) was slightly misleading because the mentioned l_B corresponded to the largest B used for each fitting. We revised the main text as follows:

(page 3. right column, 3rd paragraph, line 17)

"For higher n_p data, we estimate $l_{SO} < 35$ nm by using a full formula (Eq. 1). Although this is comparable to l_B at the largest B used for each fitting, data points at lower B (i.e., larger l_B) are more important for WAL/WL due to their steep dependence on B , making the fitting based on Eq. 2 robust. Indeed, the fit based on Eq. 2 reproduces the low-field part of the experimental MC data perfectly (Fig. 4e-o), underscoring that the signal originates from WAL as described by Eq. 2."